# Cannibalism makes invasive comb jelly, *Mnemiopsis leidyi*, resilient to unfavourable conditions

Jamileh Javidpour[1✉], Juan-Carlos Molinero[2,6], Eduardo Ramírez-Romero[3,6], Patrick Roberts[4,5] & Thomas Larsen[4✉]

The proliferation of invasive marine species is often explained by a lack of predators and opportunistic life history traits. For the invasive comb jelly *Mnemiopsis leidyi*, it has remained unclear how this now widely distributed species is able to overcome long periods of low food availability, particularly in their northernmost exotic habitats in Eurasia. Based on both field and laboratory evidence, we show that adult comb jellies in the western Baltic Sea continue building up their nutrient reserves after emptying the prey field through a shift to cannibalizing their own larvae. We argue, that by creating massive late summer blooms, the population can efficiently empty the prey field, outcompete intraguild competitors, and use the bloom events to build nutrient reserves for critical periods of prey scarcity. Our finding that cannibalism makes a species with typical opportunistic traits more resilient to environmental fluctuations is important for devising more effective conservation strategies.

[1] Department of Biology, University of Southern Denmark, Campusvej 55, 5230 Odense M, Denmark. [2] MARBEC-IRD/CNRS/IFREMER/Univ Montpellier, Avenue Jean Monnet, BP 171, 34203 Sète Cedex, France. [3] Fish Ecology Group, Instituto Mediterráneo de Estudios Avanzados, IMEDEA (CSIC-UIB), C/ Miquel Marqués 21, 07190 Esporles, Illes Balears, Spain. [4] Max Planck Institute for the Science of Human History, Kahlaische Str. 10, 07745 Jena, Germany. [5] School of Social Sciences, The University of Queensland, St Lucia, QLD 4072, Australia. [6]These authors contributed equally: Juan-Carlos Molinero, Eduardo Ramírez-Romero. ✉email: jamileh@biology.sdu.dk; larsen@shh.mpg.de

In a world with rapidly changing ecosystems and re-distributions of biodiversity, it is becoming increasingly important to understand how traits and adaptions allow certain species to colonize or even dominate new habitats. The success of exotic, invasive species is often broadly ascribed to a lack of natural predators or built-in competitive advantages such as voracious eating and high reproductive output[1]. However, such opportunistic traits also make newly established large populations of exotic species vulnerable to extinction during sudden changes in environmental conditions and food availability[2]. If we are to predict the long-term impacts of exotic species on local ecosystems, especially for invasive species with demonstrable environmental and economic impacts, it is crucial to explore the mechanisms by which these species overcome periodic food scarcity and environmental stress. Understanding these behaviors and adaptations is important in order to model the expansion of these taxa and develop more appropriate species-specific management strategies[3,4].

The invasive comb jelly *Mnemiopsis leidyi* A. Agassiz, 1865 is an exemplar of a widespread and prolific marine invasive predator with a variety of opportunistic traits including bloom-and-bust population dynamics and rapid growth[1,5–9]. As these jellies compete with fish and fish larvae they can cause cascading effect on crucial planktonic food webs and disrupt commercial fisheries[10,11]. A remarkable aspect of the spread of this species across Eurasian waters is that it is now distributed in habitats with much longer low-feeding periods (58°N) than in its native habitat along the Atlantic coast of North America (42°N)[12]. Yet, *M. leidyi* is holoplanktonic with no known benthic resting eggs, cysts, or specialized overwintering stages[13]. Given that *M. leidyi* is a fast shrinking species with a low reserve capacity[14], it is critical that the adult population maintains its nutrient reserves during late season. Surprisingly, however, adults appear to invest massively in reproduction at this time despite no or low survival chances of their progeny[6,8]. Larvae stop growing, shrink in size[15] and die after three weeks of full starvation[16]. In *M. leidyi*'s northernmost invasive habitats, there is a clear absence of larvae in winter and spring[17]. Hence, high reproductive investments during late season seems counterproductive, yet *M. leidyi* populations are able to survive most winters within their current range in northern Eurasian waters[18,19].

Here, we test the hypothesis that *M. leidyi* periodically resort to cannibalism to meet nutritional needs[20]. Cannibalism has been largely ignored as a putative strategy for nutritional supplementation in studies of gelatinous zooplankton, both in native and exotic habitats[21–23]. We carried out high frequency field observations of both prey and predators covering pre-bloom to post-bloom periods of *M. leidyi* in the south-western Baltic Sea (see map in Fig. 1), from August to October, to estimate ingestion rates relative to availability of prey and to understand how abiotic and biotic factors affect the population dynamics of *M. leidyi*. To investigate whether cannibalism only occurs under special conditions in the field, we also incubated *M. leidyi* adults and larvae together in the laboratory using stable isotope labeling to track the ingestion of larvae. Both our laboratory and field results support that adult *M. leidyi* cannibalize their larvae. A shift from interspecific to intraspecific predation in late summer allowed the adult population to build its nutrient reserves during a period where their basal metabolism is very high. We posit that this strategy confers a fitness advantage in regions with long and cold winters, and that cannibalism may be a key trait behind *M. leidyi*'s success in establishing permanent populations in its northernmost exotic habitats.

## Results

**Seasonal population growth.** We carried out a daily high frequency sampling of *M. leidyi* and their prey in Kiel Fjord (Fig. 1) in August and September 2008, the period before and after bloom collapse. Abundance and total specimen size of *M. leidyi* peaked around the 245th day of the year (September 1, Fig. 2A, B), seven days after the abundance peak of mesozooplankton (copepods), the main prey for adults, but ten days before the peak of microzooplankton (tintinnids, ciliates and dinoflagelates), the main prey for larval *M. leidyi* (Fig. 2D). We did not observe an increase in the relative abundance of larvae to adults in response to increased microzooplankton availability (Fig. 2C, D). Assuming that *M. leidyi* needs more than 20–25 µg copepod C ind$^{-1}$ per litre to support population growth[15], prey abundance was sufficient to support exponential growth of the *M. leidyi* population until it peaked around the 245th day of the year. During the growth and collapse phases, the relationship between the adult and larval *M. leidyi* abundances followed two different trajectories (Fig. 3). While the abundance relationship between *M. leidyi* adults and larvae was linear during the growth phase, adult abundances increased during the phase when both juvenile *M. leidyi* and zooplankton prey abundances collapsed (Figs. 3 and 2C).

**Direct evidence of cannibalistic behavior.** To confirm the potential for cannibalism under laboratory conditions, we incubated adult *M. leidyi* together with $^{15}$N enriched *M. leidyi* larvae in September 2016 (see $^{15}$N concentrations in Table 1). After 36 h of incubation, the $^{15}$N concentrations of adults fed $^{15}$N enriched larvae were significantly higher than adults from the control treatment ($t_{3.3} = -4.96$, $P = 0.008$; Table 1). In terms of biomass gain from feeding on larvae, the consumed larvae provided $4.1 \pm 0.1\%$ carbon and $2.5 \pm 0.1\%$ nitrogen ($n = 3$) of the total elemental content of each adult (see Eqs. 1–2 in

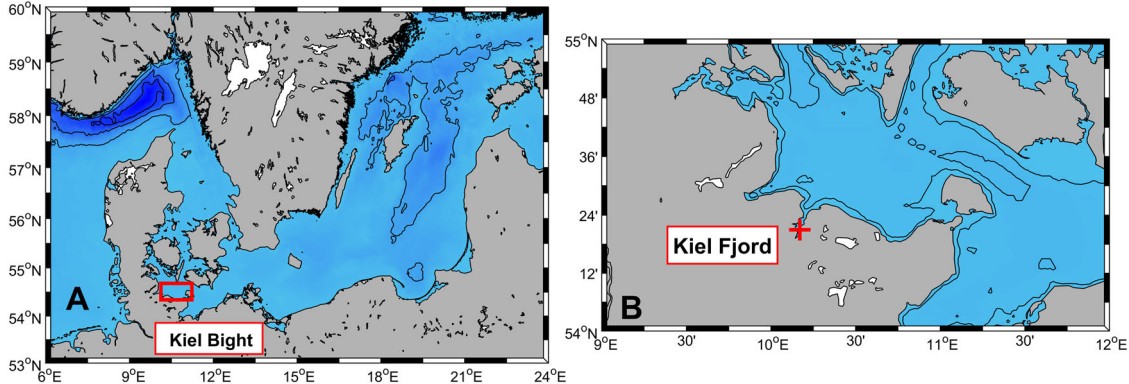

**Fig. 1 *Mnemiopsis leidyi* and their prey were sampled in Kiel Fjord.** which is a 17 km long inlet of the south-western Baltic Sea.

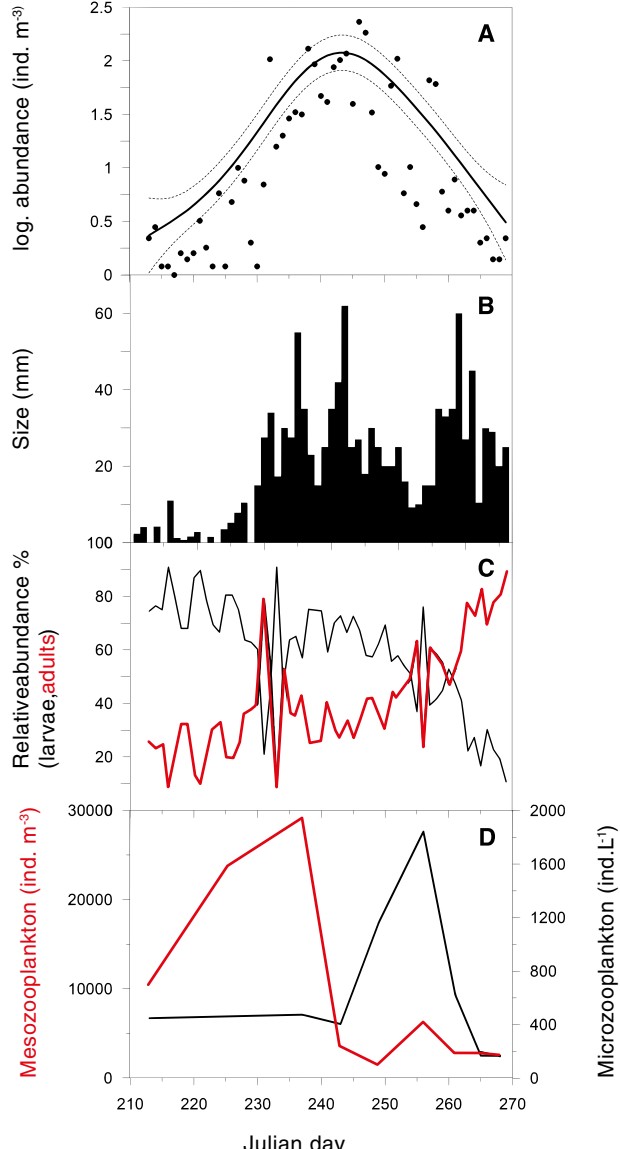

**Fig. 2 Population data of *Mnemiopsis leidyi* and their zooplankton prey from August to October 2008 in Kiel Fjord.** The *x*-axis denotes the sampling period in Julian days from August 12 to October 21, 2008. **a** Daily abundance variations of *M. leidyi*, (**b**) cumulated length of specimens collected each day (6094 specimens collected across the entire sampling period), (**c**) relative abundance of *M. leidyi* larvae and adults (large developmental stage), and (**d**) the prey field based on weekly samplings, as represented by the abundance of mesozooplankton and microzooplankton.

Material and Methods). The discrepancy between carbon and nitrogen fractions can be explained by the higher C:N ratio of the larvae than the adults (Table 1). In the control treatment without adults, we detected zero larval mortality during the 36 h incubation.

To obtain evidence of cannibalistic behavior in the natural environment, we photographed field sampled adults within 30 min after sampling. These photographs were taken in the laboratory of GEOMAR during the *M. leidyi* bloom collapse in September 2008. The photographs show two larvae inside the auricles of an adult (Supplementary Fig. 1). We rule out post-capture larval ingestion because the mesh size for capturing the adults was too big for larvae.

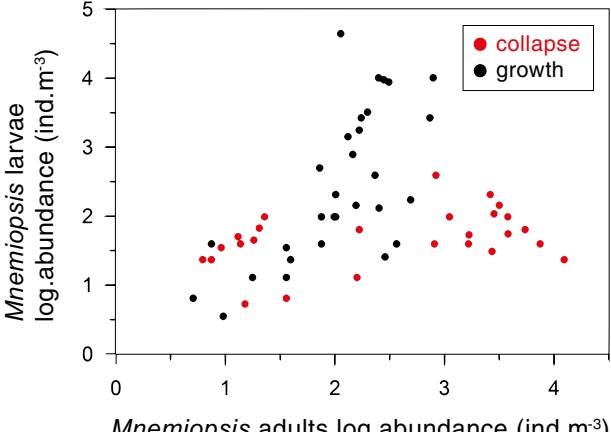

**Fig. 3 Density dependence relationship between the relative abundances of *Mnemiopsis leidyi* larvae and adults from August to October 2008 in Kiel Fjord.** Larval abundances were highest at average adult abundances, and lowest when adult abundances were either small or large.

**Population development and drivers**. According to our estimates of ingestion rates (see Eqs. 3–7), the shift from predominantly interspecific to intraspecific predation decreased the daily rations of adults. However, the rations during this post-bloom phase remained sufficiently high, around 10–20% body carbon d$^{-1}$, to sustain the adult population for an additional 2–3 weeks (Figs. 2 and 4). By comparison, predation on copepods yielded daily rations of up to 50% of body carbon d$^{-1}$ during the peak of the copepod bloom, day 237 (Fig. 4), which is in line with former observations[24]. See Supplementary Table 1 for mesozoo-plankton and copepods assemblages during late summer 2008 in Kiel Fjord. We employed Structural Equation Modeling (SEM) because it identifies both direct and indirect drivers of *M. leidyi* seasonal population growth, as well as their relative importance. The SEM results showed that increased temperature and food availability, i.e., microzooplankton abundance, were the leading drivers of *M. leidyi* population growth (Fig. 5). For both factors, we found positive effects, which are represented by the path coefficients 0.51 and 0.47, respectively. By contrast, the bloom decline was primarily associated with decreased temperature and cannibalism (path coefficients: 0.34 and −0.43), respectively. Our observational data show that the copepod supply was depleted soon after the copepod population peaked and that *M. leidyi* adults then shifted to feed on their larvae.

## Discussion

To our knowledge, we have presented the first unequivocal evidence that adult *M. leidyi* cannibalize their own larvae. This finding fills an important gap in knowledge as to how an invasive species exhibiting boom-and-bust behavior is able to survive long periods of nutrient scarcity. By shifting to cannibalizing their own larvae after emptying the prey field, adults can continue their growth. This behavior provides *M. leidyi* with the possibility of outcompeting intraguild species (e.g., *Pleurobrachia pileus*) by feeding on a wide size range of prey and, at the same time, enables it to build up nutrient reserves under unfavorable conditions. *M. leidyi* populations are otherwise vulnerable to local extinctions during this period because typically-sized adults have energy reserves for 9 days at 20 °C[25]. Since basal metabolic rates are reduced exponentially with temperature, adults have energy reserves for up to 80 days at 3 °C[25]. Hence, provided that *M. leidyi* adults maintain their biomass well after the bloom collapse, they have sufficient reserves to survive long periods of low feeding under cold water temperatures. The fact that *M. leidyi* larvae function as nutrient and energy reserves may provide an explanation

 3

**Table 1 Size, elemental content and atom % $^{15}$N of adult and larval _Mnemiopsis leidyi_ in the feeding experiment.**

| Treatment | Size (mm) | Carbon (mg) | Nitrogen (mg) | C:N | Atom % $^{15}$N |
|---|---|---|---|---|---|
| Adult control | 22.0 ± 1.0 | 0.89 ± 0.08 | 0.22 ± 0.07 | 3.8 ± 0.7 | 0.3679 ± 0.0008 |
| Adult labeled | 22.3 ± 1.1 | 0.91 ± 0.09 | 0.21 ± 0.01 | 4.3 ± 0.2 | 0.3728 ± 0.0020 |
| Larvae labeled | 4.5 ± 0.7 | 0.037 ± 0.004 | $5.3 \times 10^{-3} \pm 4.0 \times 10^{-4}$ | 6.9 ± 0.1 | 0.5579 ± 0.0826 |

The numbers signify mean and standard deviation values across three replicates in each treatment. For the adults, the carbon content was estimated from the size of the adults (see Eq. 1), and the nitrogen content was estimated from the C:N ratio (atomic). For the larvae, the carbon and nitrogen contents represent the average amount assimilated by the adults in each treatment during the 36 h incubation (see Eqs. 1 and 2). The consumed larvae provided 4.1 ± 0.1% carbon and 2.5 ± 0.1% nitrogen ($n = 3$) of the total elemental content of each adult. The discrepancy between carbon and nitrogen values can be explained by the higher C:N ratio of the larvae than the adults.

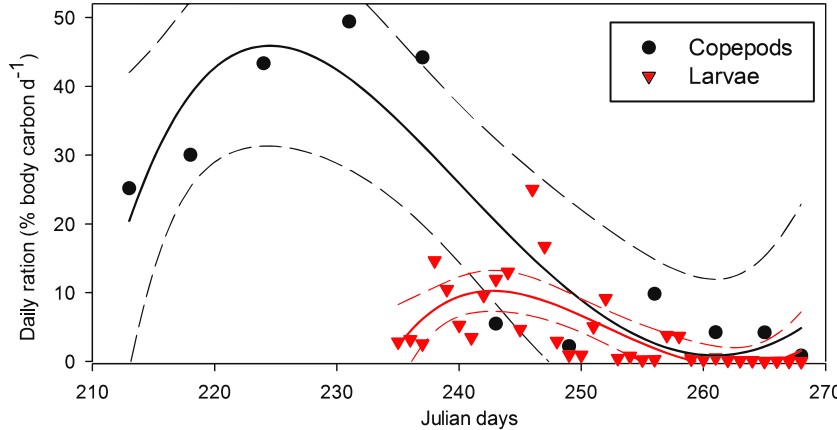

**Fig. 4 Daily ration (% body carbon d$^{-1}$) of the _Mnemiopsis leidyi_ adults preying on mesozooplankton (copepods; black filled circles) and _M. leidyi_ larvae (red filled triangles) using Eqs. 6 and 7.** The regression curves (solid lines) are based on inverse cubic polynomial fits with 95% confidence intervals (dashed lines).

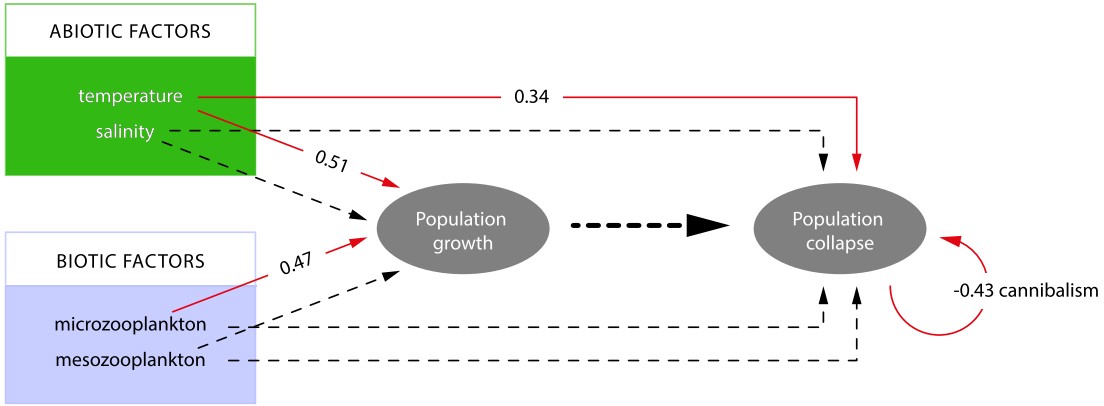

**Fig. 5 Path diagram showing how abiotic factors, prey availability, and cannibalism control the population of _Mnemiopsis leidyi_.** Solid red paths are statistically significant ($p < 0.05$), whereas black dashed lines are not. The standard coefficients are displayed at each significant path. Both temperature coefficients are positive because the bloom and collapse phases are concurrent with temperature increase and decrease, respectively.

as to why some bloom forming species are well adapted to dramatic population fluctuations in contrast to other prolific species that appear to become more vulnerable to extinction under increasingly variable conditions[2].

Cannibalistic behavior in animals often adheres to a common set of rules: juveniles are eaten more frequently than adults; the behavior is concurrent with a decrease in alternative forms of nutrition; and a decrease in population density directs intraspecific prey to the fittest individuals during times of food scarcity[21,22]. While cannibalistic behavior is a common ecological phenomenon in the animal kingdom[21,22,26], our study of

_M. leidyi_ is rare in that cannibalism becomes detectable after the total collapse of the copepod, i.e., the prey, population, ruling out the possibility that adults and juveniles were competing for prey. In fact, adult _M. leidyi_ may even enhance food availability for their larvae by consuming microzooplankton predators[27]. We argue that the population dynamics between adults and their larvae necessitates the need for ecologists and conservationists to study _M. leidyi_ populations as a coherent whole—an intergenerational, multicellular organism. By investing massively in reproduction during late summer _M. leidyi_ increases its ability to empty the prey-field across different size classes and built up

nutrient reserves. Since larvae cannot survive winters in the species' northernmost habitats, our study suggests that the primary purpose of M. leidyi larvae is to gather and store energy and nutrients for adults. This strategy is akin to autophagy within multicellular organisms during starvation periods, the process through which many insects and mammals use fat bodies as nutrient and energy reserves for overwintering[28,29]. However, in order to fully understand the relative costs of adults to invest in reproduction versus the energy they gain from larval cannibalism, it would be necessary to characterize the costs of egg production for field sized adults (~20 mm length), as well as the energy larvae gain by predating small size zooplankton.

The identification of this strategy has major implications for understanding the adaptive capacities of M. leidyi and designing appropriate conservation strategies that can control its spread. In the Black Sea, the introduction of the native predator Beroe ovate in the late 1990s decreased absolute M. leidyi abundance. Nevertheless, the efficiency of M. leidyi's prey capture technique has meant that its impact on interspecies zooplankton continues to be substantial during late summer bloom periods[30]. Similarly, it is likely that eutrophication and over-fishing of zooplanktivorous fish prior to M. leidyi arrival in the Black Sea exacerbated the ferocity of boom-and-bust population dynamics of this invasive species[6]. A range of measures such as decreasing eutrophication and commercial fishing of forage fishes (to increase intraguild competition) should be considered to curb M. leidyi's dominance and its ability to establish permanent populations within its current northernmost invasive range[31]. Our dataset also has significant implications for M. leidyi dynamics within its native range. Many of the environmental conditions that have made Eurasian locations physiologically favorable for M. leidyi invasion, warming temperatures and increased eutrophication, also exist in the species' native areas. Increasing disturbance of marine environments suggests that the negative impact of M. leidyi on ecosystem services may also become an increasing problem in the species' native habitats as it is highly likely that it can turn its cannibalistic boom-and-bust strategies on and off in changing environmental circumstances[18,32,33].

As well as further documenting the ubiquity of cannibalism in the animal kingdom and its far-reaching consequences for native and exotic ecologies, our dataset also provides some hints regarding the evolutionary origins of cannibalism—whether it is a product of convergent evolution or a basic metazoan trait[26,34]. The earliest evidence for metazoan cannibalism can be traced to the Cambrian period where one specimen of Ottoia, a priapulid, had a proboscis of another Ottoia preserved in its gut[35,36]. Cannibalism may even have played a role in the evolutionary transition from single cell to multicellular organisms because sponges, Porifera, have specialized cells that eat other cells in times of food scarcity[37]. Here, we demonstrate that cannibalism is an important trait for a member of Ctenophora, but whether it is a basic or derived metazoan trait remains an open question until more conclusive evidence of cannibalism or lack hereof can be obtained from other Ctenophora taxa. Hence, more research characterizing the role of cannibalism in a comparative context is essential for understanding the life history strategies of ctenophores and for the evolutionary origins of cannibalism.

Characterizing life history strategies of ctenophores is crucial for understanding the origins of metazoan traits, as well as for developing more accurate predictions of the future economic and ecological impacts of prolific species. M. leidyi is probably the most-studied ctenophore genus in the world because of its great abundance next to densely populated areas in its native habitats, and because of its widespread impact on zooplankton and ichthyoplankton following its invasion of Eurasian waters. Our study suggests that cannibalism is vital for M. leidyi adults to overcome prey scarcity during a critical period when its basal metabolic rate is high. Without this strategy, M. leidyi would be more likely to become locally extinct during the long and cold winters in its exotic habitats. The extent to which this behavior confers a fitness advantage on M. leidyi adults to invest in reproductive over somatic growth under different conditions requires further parametrization in terms of interspecific predation, exploitation of different prey fields, and costs of egg production[14,38]. Given the increasing disturbance of marine environments and spread of exotic species, our finding is important for devising more effective conservation strategies. Our data encourage ecologists and conservationists to compare individual adaptive traits across the population structure of a given species in order to explore how this might contribute to its range expansion and its impact upon other competing taxa. Furthermore, our findings highlight cannibalism as a basal, but variably expressed, trait of the animal kingdom that can increase fitness and adaptability within a variety of demographic and environmental contexts.

## Methods

**Field sampling.** Plankton sampling was performed at the GEOMAR deck (54° 19′48′′N, 10°9′1′′E, see map in Fig. 1) between 10.00 and 11.00 h from Mondays to Saturdays in the period of August 12 to October 21, 2008. Samples of M. leidyi were taken with a WP2 net (0.8 m net opening, 500 μm mesh size), with three vertical hauls being made at each sampling occasion from the bottom (6 m) to the surface. Individuals were counted and measured alive immediately after sampling since M. leidyi disintegrate in standard fixation solutions. Total length was measured to the nearest 0.1 mm for individuals with closed lobes. Mesozooplankton were sampled at the same station at weekly intervals by using a plankton net (0.6 m diameter opening, 200 μm mesh size) from integrated vertical tows of 6 m depth to the surface. Samples were preserved in 5% buffered formalin-seawater mixture for later quantification. Mesozooplankton samples were divided using a plankton splitter[39] until at least 100 individuals (including copepodites and adults, excluding nauplii) of the numerically dominant copepod species were found in a single subsample. All mesozooplankton specimens in the samples were identified at least to genus level under a dissecting microscope by Paulsen, Hammer, Malzahn, Polte, von Dorrien and Clemmesen[40] using the taxonomic guide by Sars[41]. See assemblages of mesozooplankton in Supplementary Fig. 2A and copepods in Supplementary Fig. 2B. For later counts of microzooplankton, water samples (250 ml) were taken from mid-depth on a weekly basis and were preserved using Acid-Lugol. Microzooplankton species composition was determined at least to the genus level using a convert microscope. Temperature and salinity were measured at a one-meter interval along the whole water column on each sampling day. Other environmental factors like wind direction, wind speed, and water level were obtained from a meteorological station at the roof of GEOMAR.

M. leidyi specimens were divided into two size categories, larvae and adults, according to their morphological features[42]. Small tentaculated cydippid larvae that had no sign of developed oral lobes and auricles and transition-stage larvae with tentacles and small oral lobes were ranked as larvae (1–9 mm). Specimens at the lobate stage and with developed auricles were counted as adults.

**Laboratory culture experiment.** To obtain direct evidence of whether adult M. leidyi consume their larvae, we performed a feeding experiment with and without [15]N labeled food. Our experiment involved three trophic levels. As food for the copepod Acartia tonsa, we first cultured the cryptophyte alga Rhodomonas sp in two different treatment levels (a) labeled cultures with F/2 medium containing labeled [15]N $NH_4NO_3$ and (b) non labeled F/2 medium. Freshly hatched nauplius of A. tonsa from GEOMAR permanent cultures were transferred to two new containers and were fed permanently with two types of Rhodomonas sp. After a month, by which time they had reached the copepodite stage of C3-4, M. leidyi larvae (size 4.6 ± 0.4 mm) were taken from the permanent cultures, kept in 20 liter buckets, and fed with two types of copepods at surplus level. Water was exchanged once a day.

All experimental organisms were kept at 15 °C, the light intensity was 100 μmol photons m$^{-2}$ s$^{-1}$ at a light:dark cycle of 16:8 h, and the salinity was 16. These conditions are typical for late summer conditions in the Kiel Fjord. For culturing Rhodomonas sp.[43] we used Provasoli's enriched seawater medium according to Thomsen and Melzner[44]. The algae were [15]N-labeled by adding 0.807 g 99 atom% [15]N-$(NH_4)_2SO_4$ (Cambridge Isotope Laboratories) and 22.011 g natural abundance $NH_4NO_3$ to 1000 ml of stem solution. 2 ml of stem solution was added to each litre of culture medium.

To be sure that the larvae accumulated a measurable amount of excess [15]N, we fed them copepods for one week. A triplicate feeding experiment was designed in two feeding levels a) M. leidyi adults were fed with labeled M. leidyi larvae and M. leidyi adults were fed with non-labeled copepods. A total of ten larvae were transferred to a 2-liter jar with one adult M. leidyi from GEOMAR continuous

culture. The adults were starved for a period of 24 h prior the experiment. The feeding experiment was terminated after 36 h. To track mortality of both *M. leidyi* larvae and copepods without predators, two extra units were added to the original design without predators. Adults and remaining larvae that were not used as feed were freeze-dried individually. All treatments were performed in triplicate.

**Isotope analyses and biomass**. For determining C and N contents, and $^{15}N/^{14}N$ ratios of the experimental *M. leidyi*, we transferred around 5 mg of *M. leidyi* dry mass into tin capsules. The samples were analyzed at Centre for Stable Isotope Research and Analysis at University of Göttingen with a Euro EA 3000 interfaced to a Delta V Plus via a Conflo IV interface. The $^{15}N/^{14}N$ ratio of atmospheric N was used as primary reference and acetanilide ($C_8H_9NO$, Merck, Darmstadt) was used for internal calibration.

To calculate the biomass of larvae assimilated by adults, we first calculated the mass of excess $^{15}N$ in adults incubated with $^{15}N$ labeled larvae:

$$15_{[N]e} = 14 + 15_{[N]_e} \times \frac{\text{atom\% } 15_{Ne} - \text{atom\% } 15_{Nc}}{100}; \tag{1}$$

where [N] signifies the mass of nitrogen for a given isotope, subscript $e$ $^{15}N$-labeled adults, and subscript $c$ $^{15}N$-unlabeled adults (the control).

Next, we estimated the biomass of larvae assimilated by adults ($j$) by rearranging Eq. 6 and substituting $e$ with $j$:

$$14 + 15_{[N]j} = 15_{[N]e} / \frac{\text{atom\% } 15_{Nj} - \text{atom\% } 15_{Nc}}{100}; \tag{2}$$

**Structural equation modeling**. SEM was used to numerically assess the complex interactions between biotic and abiotic drivers of *M. leidyi* abundance and partition the direct and indirect effects of environmental drivers on *M. leidyi* seasonal growth. To partition the net effects of environmental variables on population growth and decline and their relative importance, data were separated into two groups, see Fig. 3, and analyzed in a framework of multigroup SEM. The first model assessed population growth and included abiotic (temperature and salinity) and biotic factors (micro and mesozooplankton). The second model assessed drivers of population collapse that besides the above abiotic and biotic factors included a density-dependent factor (adult abundance) with and without cannibalism. Based on previous observations[17], we hypothesized that (i) *M. leidyi* population growth is driven by food availability, and the match with warming condition, (ii) while the population collapse may response to food depletion, temperature decline and cannibalism.

**Daily ration assessment**. To estimate the mean community carbon content per individual copepod, copepod abundances were transformed to micrograms of carbon using carbon contents from Supplementary Table 1. Clearance rates for adults ($Cl_{ad}$) were obtained from Granhag, Møller and Hansson[45] including a temperature regulation via $Q_{10}$. For copepod predation (copepod *Oithona* sp., size ca. 0.45 mm), this rate was used:

$$Cl_{cop}(\text{ind}^{-1} L\, h^{-1}) = 0.0054 \times l^{2.01} \times Q_{10}^{\frac{\text{Temp}-20}{10}}. \tag{3}$$

In the case of larvae predation, after copepod depletion (day 237), this equation was applied (gelatinous zooplankton *Oikopleura dioica*, 1.5 mm):

$$Cl_{larv}(\text{ind}^{-1} L\, h^{-1}) = 0.0012 \times l(\text{mm})^{2.54} \times Q_{10}^{\frac{\text{Temp}-20}{10}}; \tag{4}$$

with $l$ (mm) oral-aboral length of adults and $Q_{10} = 2.7$[46].

The daily ration of adults was defined as the ingestion rate *per capita* ($DR$, % body C d$^{-1}$), feeding on larvae ($DR_{larv}$) or copepods ($DR_{cop}$) (Eqs. 4 and 5). The carbon content per adult ($CC_{ad}$) was defined as a function of $l$ according to Sullivan and Gifford[42]. Larvae carbon content ($CC_{larv}$) is defined in Table 1 (0.037 mg C):

$$CC(\text{mg C ind}^{-1}) = 0.0017 l^{2.0138}; \tag{5}$$

and then, ingestion ($I$) by adults ($CC_{ad}$) of copepods ($CC_{cop}$) or larvae ($CC_{larv}$) predation was defined as:

$$DR_{larv}(\% \text{ body C d}^{-1}) = \alpha_l \times Cl_{larv} \times CC_{larv} \times A_{larv}/CC_{ad} \times 100; \tag{6}$$

$$DR_{cop}(\% \text{ body C d}^{-1}) = \alpha_c \times Cl_{cop} \times CC_{cop} \times A_{cop}/CC_{ad} \times 100; \tag{7}$$

where $A_{larv}$ and $A_{cop}$ are the larvae and copepod abundances (individuals m$^{-3}$) and an assimilation efficiency $\alpha_l = 0.8$ and $\alpha_c = 0.4$[47]. The carbon content of copepods was set to 0.9 µg C ind$^{-1}$, following in situ copepod assemblages (Supplementary Fig. 3).

**Statistics and reproducibility**. The Baltic Sea map (Fig. 1) was created with the m_map package for Matlab R2018[48]. The Independent Samples T-Test for testing isotopic differences between the two treatments ($n = 3$) was performed in R version 3.6.1. The SEMs were run in AMOS (version 21). All SEM data were equally weighted and standardized to zero mean and unit variance. SEM was applied on a matrix of abiotic (temperature, salinity) and biotic (microzooplankton, mesozooplankton and cannibalism) factors. The strength and sign of the links and

quantification of the SEM were determined by simple and partial multivariate regression and Monte Carlo permutation tests ($n = 1000$), whereas chi-square values were used to assess the fit of the overall path model. The individual path coefficients, i.e., the partial regression coefficients, indicate the relationship between the causal and response variables. Significance levels for individual paths between variables were set at $p = 0.05$. To analyze the data for SE Ms, we compared models with the observed covariance matrix, using maximum likelihood and $\chi^2$ as goodness-of-fit measures. When $P < 0.05$ data were considered significantly different from the model. As data from the individual groups fit the model ($P > 0.05$), we considered legitimate to perform a multigroup SEM analysis. Significance levels for individual paths between variables were set at $\alpha = 0.05$. The daily ration assessments were generated in Matlab R2018, and the results were plotted with Sigmaplot v.14. No specific code was developed for Fig. 4.

**Reporting summary**. Further information on research design is available in the Nature Research Reporting Summary linked to this article.

## Data availability
The datasets generated during the current study are deposited in PANGAEA (https://doi.org/10.1594/PANGAEA.893355) or available upon request from the corresponding author, Dr. Jamileh Javidpour.

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

## Acknowledgements

J.J. and J.-C.M. were supported by the Horizon 2020 project GoJelly no 774499; J.J. by the Deutsche Forschungsgemeinschaft (DFG) grant no. JA 2008/1-1; J.-C.M. by the European Commission (OCEAN-CERTAIN, FP7-ENV-2013-6.1-1; no: 603773); T.L. and P.R. by the LOMVIA project (FKZ 03F0805A), part of the Changing Arctic Ocean programme funded by the German Federal Ministry of Education and Research (BMBF), and the Max Planck Society; E.R.-R. by "Programa Vicenç Mut-Govern de les Illes Balears- Conselleria d'Innovació, Recerca i Turisme". We thank Drs. Clemmesen and Paulsen for sharing their original data on zooplankton species composition.

## Author contributions

J.J. and J.-C.M. conceived and conducted the field survey. J.J. and T.L. designed and carried out the feeding experiment. J.-C.M. and E.R.-R. analyzed the data. T.L. and R.P. wrote the paper with input from J.J., J.-C.M., E.R.-R and final approval from all authors.

## Competing interests

The authors declare no competing interests.
