## [Peer Review File · Communications Biology]

Reviewers' comments:

Reviewer #1 (Remarks to the Author):

This paper shows how cannibalism by adult invasive ctenophores on their larvae could increase adult survival under low food conditions by having the larvae act as energy reserves. I think this is a very interesting theory, and it certainly seems like a plausible strategy. The stable isotope experiment convincingly shows that cannibalism occurs. Also the metabolic rates at low temperatures are useful and novel data. However, I am not yet convinced by the daily ration estimates, because quite low individual carbon content values are used for copepods. Using these values, mesozooplankton food availability quickly becomes limiting but how representative are these values to the field conditions? Considering data is available on in situ mesozooplankton composition, this should be discussed in more detail. In general the study is well written. It makes an interesting and novel addition and I think it should be published in a revision where below mentioned suggestions are addressed.

Timing: This paper combines field sampling of an invasive ctenophores population in 2008 with a feeding trial experiment carried out in 2016. This long time between different parts of the study does not have to be an issue, but it would be interesting to read on why this is the case, which new insights triggered the experiment? In the paper, please mention that the experiment was carried out in 2016 and on which dates.

Salinity is reported both in g kg and in PSU, please stick to a single unit and do not write "22 PSU" but "a salinity of 22"

The sampling period is very short, only two months, but in this period sampling frequency was high. Still, do you have, or can you cite, data that shows size distribution of *M. leidyi* in Kiel Fjord or other temperate invaded areas in winter/spring, showing absence of larvae in winter and spring is really a thing?

Prey is called both "mesozooplankton" and "copepods". Does mesozooplankton also include other groups? I think the description of the available prey, just saying it was "copepods" is not enough. Also considering the next comment.

My main issue with the paper is this: how sensitive are your daily ration estimates to the choice of copepods? You use an individual carbon content value of *Oithona*, a small copepod, of 0.5 ugC/ind, why is this? Was this the dominant species present? Why not use *A. tonsa* values which are 5/10 times higher (Jaspers et al 2017), not to mention other copepods like *Temora* which are even bigger. A large part of the paper and its conclusions are based on the assumption that mesozooplankton daily ration becomes insufficient, but how sensitive is this to this assumption of low carbon content for the prey items. Is it still the case when *Acartia* values are used for copepods?

A useful addition might be the estimation of the amount of energy invested into reproduction-for-cannibalism, compared to the amount of energy gained from larval cannibalism. Isn't it more profitable to directly invest energy into reserves, rather than reproduction?

Discussion correctly highlights the impact of overfishing and eutrophication on *M. leidyi* proliferation, rather than blaming the comb jelly for everything.

It would be interesting to discuss the work of Augustin et al. 2014 who discuss that *M. leidyi* has a very low reserve capacity, this suggests that "storing" reserves as larvae might thus be a feasible and needed strategy for survival.

Additionally, this “larval cannibalism” strategy seems to be complementary to the finding that adult *M. leidyi* enhance food availability for their larvae by consuming microzooplankton predators (McNamara et al. 2013).

In methods you describe delta15N values but these are not used in the paper, only the raw 15N values?

Fig. 2B shows average size?

I think your fig. 2 D right axis should be “microzooplankton”?

References

Augustine, S., Jaspers, C., Kooijman, S.A.L.M., Carlotti, F., Poggiale, J.-C., Freitas, V., Van der Veer, H., Van Walraven, L., 2014. Mechanisms behind the metabolic flexibility of an invasive comb jelly. *J. Sea Res.* 94, 156–165.

Jaspers, C., Costello, J., Sutherland, K., Gemmell, B., Lucas, K., Tackett, J., Dodge, K., Colin, S., 2017. Resilience in moving water: Effects of turbulence on the predatory impact of the lobate ctenophore *Mnemiopsis leidyi*: *Mnemiopsis leidyi* feeding in turbulence. *Limnology and Oceanography* 63. <https://doi.org/10.1002/lno.10642>

McNamara, M.E., Lonsdale, D.J., Cerrato, R.M., 2013. Top-down control of mesozooplankton by adult *Mnemiopsis leidyi* influences microplankton abundance and composition enhancing prey conditions for larval ctenophores. *Estuar. Coast. Shelf S.* 133, 2–10.

Reviewer #2 (Remarks to the Author):

Javidpour et al. document cannibalism in the ctenophore *Mnemiopsis leidyi* in lab experiments. This is a novel report for the phylum (to my knowledge) and should be of broad interest. More importantly, the authors then make the reasonable conjecture that similar cannibalism happens in the Baltic Sea and that this behavior can explain the ability of the species to survive winter outside its native range with greater energy reserves. This type of experimental work is outside my area of expertise, but they appear sound and compelling. Further, that these experiments provide positive evidence that cannibalism is an important strategy allowing individuals of this ecologically important species to build energy stores to survive winter conditions appears reasonable.

For my taste, however, I think the conclusive language is a bit strong. They have not shown that cannibalism happens in natural conditions, either in the Baltic or in the native range of the species. It seems obvious that this behavior must have evolved in different conditions than those in the Baltic. I'd like to see the authors introduce a bit more discussion on what relevant knowledge is missing in understanding this phenomenon, rather than being so strong in concluding that they have uncovered a “crucial strategy for *M. leidyi* adults to overcome prey scarcity” outside its native habitat. Is the strategy potentially crucial within its native habitat? That said, this knowledge is crucial to understanding the present and potential impact of this invasive species, as the authors rightly point out.

Finally, I find the evolutionary discussion of cannibalism, trying to capitalize on the status of Ctenophora as potentially the earliest diverging metazoan lineage to be entirely unconvincing. It is forced and really adds little to the overall understanding of the evolution of cannibalism. This behavior

exists within microbial eukaryotes and is widespread across different metazoan lineages. An alternative explanation is that sometimes ecological conditions favor the evolution of cannibalism and that it is a labile trait. Given that *Mnemiopsis* is a highly derived ctenophore (see any phylogeny of Ctenophora paper) and one whose biology (being brackish, etc.) is unusual within the phylum, it seems more likely that cannibalism is a derived trait in this instance.

In summary, this manuscript needs little revision to become published as a paper that will attract wide interest. It documents cannibalism for the first time within Ctenophora, it shows how this behavior likely benefits the species by creating energy stores that allow it to survive winter, and it correctly notes that this information is crucial for marine managers needing to understand its persistence and possible spread to other regions.

Reviewer #3 (Remarks to the Author):

I believe this paper offers a new, interesting potential explanation for the success that the ctenophore *mnemiopsis leidyi* is having by colonizing new ecosystems. The paper is well written (except for few little misuses of English, likely due to the fact that none of the authors is a native speaker) and integrates several data to support the conclusion that the ctenophore is able to feed on its own larvae and this is the key of its success. However, this conclusion is based on data collected during only one bloom and adult ctenophores have never been seen while feeding on their larvae nor gut content analyses indicated that larvae were in the gut of adults. Therefore I think the authors should present cannibalism as a potential explanation for the success of the ctenophore and not as "the" explanation. For example, at the beginning of the Discussion they state this paper provides the first "evidence of cannibalism in *M. leidyi*, but I do not see eviudedence, but data that may support it. I suggest the suthors to slightly change the overall tone of the ms in this perspective.

Reviewer #1:

#1 This paper shows how cannibalism by adult invasive ctenophores on their larvae could increase adult survival under low food conditions by having the larvae act as energy reserves. I think this is a very interesting theory, and it certainly seems like a plausible strategy. The stable isotope experiment convincingly shows that cannibalism occurs. Also the metabolic rates at low temperatures are useful and novel data. However, I am not yet convinced by the daily ration estimates, because quite low individual carbon content values are used for copepods. Using these values, mesozooplankton food availability quickly becomes limiting but how representative are these values to the field conditions? Considering data is available on in situ mesozooplankton composition, this should be discussed in more detail. In general the study is well written. It makes an interesting and novel addition and I think it should be published in a revision where below mentioned suggestions are addressed.

REPLY: We thank the Reviewer for their positive comments in relation to the interesting information and theoretical background we present in our paper, as well as their enthusiasm for its publication. We are also happy that they agree that our experiment is convincing and deem this and the metabolic rates to provide useful and novel data. In relation to the daily ration estimates, we have now re-evaluated our approach following their informative comments. While we previously based our estimate on the most abundant species, *Oithona*, we now base our estimates on a representative assemblage of mesozooplankton, i.e. our revised values are now representative of field conditions. For details to how and why we made this change, please see our reply further down. In short, we changed from 0.5 to 0.9 $\mu\text{g C}$ / zooplankton specimen and the daily ration is now 50% instead of 30%. This change does not have any impact on the clear overall importance of cannibalism on larvae.

#1: Timing: This paper combines field sampling of an invasive ctenophores population in 2008 with a feeding trial experiment carried out in 2016. This long time between different parts of the study does not have to be an issue, but it would be interesting to read on why this is the case, which new insights triggered the experiment? In the paper, please mention that the experiment was carried out in 2016 and on which dates.

REPLY: We have now added that the experiment was carried out in 2016 to the main text as well as the dates. Yes, the publication has been a long time in the making. The study started in 2008 as a Bachelor project supervised by Javidpour. The initial assessment of the high frequency field sampling data suggested cannibalism, but lack of resources after the sampling hampered further data analysis until 2015 when Molinero and Ramirez-Romero modelled prey-predator dynamics. These models showed that cannibalism was the only parsimonious explanation for the peculiar population dynamics, but we still lacked direct evidence. For this reason, Javidpour contacted Larsen (both were in Kiel at the time) to discuss an isotope labeling experiment, which took place in September 2016. As an additional line of evidence, Javidpour also obtained direct visual of cannibalism from field collected specimens. We hope that this provides a 'historical' summary to the background of this paper and shows that it was designed and undertaken in a step-wise, logical fashion.

#1: Salinity is reported both in g kg and in PSU, please stick to a single unit and do not write “22 PSU” but “a salinity of 22”

REPLY: We have now altered as suggested. We now report salinity without units according to the traditions in Oceanography: <https://www.nature.com/scitable/knowledge/library/key-physical-variables-in-the-ocean-temperature-102805293/>

#1: The sampling period is very short, only two months, but in this period sampling frequency was high. Still, do you have, or can you cite, data that shows size distribution of *M. leidyi* in Kiel Fjord or other temperate invaded areas in winter/spring, showing absence of larvae in winter and spring is really a thing?

REPLY: We thank the reviewer for this point. Citing Javidpour *et al.* (2009), we have now added an additional short sentence (lines 60-61) that mentions that a clear absence of larvae in winter and spring in Kiel Fjord.

#1: Prey is called both “mesozooplankton” and “copepods”. Does mesozooplankton also include other groups? I think the description of the available prey, just saying it was “copepods” is not enough. Also considering the next comment.

REPLY: Copepods were the most abundant (90%) group of mesozooplankton in Kiel Fjord, but not the only one. Hence, we are now using the term mesozooplankton throughout. That said, we still base our daily ration estimates on copepods but now on all the identified species rather than just one. We added this to the Discussion (lines xx), but also read our reply below for more details.

#1: My main issue with the paper is this: how sensitive are your daily ration estimates to the choice of copepods? You use an individual carbon content value of *Oithona*, a small copepod, of 0.5 ugC/ind, why is this? Was this the dominant species present? Why not use *A. tonsa* values which are 5/10 times higher (Jaspers et al 2017), not to mention other copepods like *Temora* which are even bigger. A large part of the paper and its conclusions are based on the assumption that mesozooplankton daily ration becomes insufficient, but how sensitive is this to this assumption of low carbon content for the prey items. Is it still the case when *Acartia* values are used for copepods?

REPLY: We understand the reviewer’s concern regarding the daily ration estimates and thank them for raising this. Following their comments, we have now sought to present a more detailed description of the mesozooplankton community based on our *in situ* sampling that was concurrent with the *M. leidyi* sampling period (day 220 to 270). We identified the non-copepods into major taxonomic groups (Fig. 1) and copepods to species level and life stage (Fig. 2; we identified three species). We have now added Figs. 1 and 2, and Table 1 shown below to SI Text 2.

Figure 1. Kiel Fjord mesozooplankton composition in 2008. The community was dominated by copepods (90%).

Figure 2. Copepod assemblages in Kiel Fjord 2008. *Pseudocalanus sp.* was the most abundant group during the bloom, followed by copepodites of *Oithona sp.* and copepods of *Acartia sp.*

Based on the copepod assemblages in Kiel Fjord 2008 and the carbon content of each species/stage (Table 1), we estimated the mean community carbon content per individual – see Fig. 3.

Table 1. Individual carbon contents of copepods.		
Group	Carbon content ($\mu\text{g C indiv}^{-1}$)	Reference
copepodites Pseudocalanus sp.	1	(Fennel 2001)
adults Acartia sp.	2.5	(Granhag, Møller & Hansson 2011)
copepodites Acartia sp.	0.8	(Jones, Flynn & Anderson 2002)
copepodites Oithona sp.	0.17	(Granhag, Møller & Hansson

		2011)
adults Oithona sp.	0.5	(Granhag, Møller & Hansson 2011)

Figure 3. The mean community carbon content per individual copepod.

The carbon content per individual is quite constant over the time. Hence, we used a mean value of 0.9 µg C indiv⁻¹ instead of the previous value of 0.5 µg C indiv⁻¹ in our daily rations estimates.

Figure 4. Daily ration estimates based on 0.9 µg C indiv⁻¹ copepod. Note that we show this figure with smoothing in the paper.

The copepod daily rations are now slightly larger with a maxima of ca. 50% instead of 30%. Using the adjusted carbon content, our results show that the population is maintained during 2-3 weeks only by

feeding on larvae, which in turn does not change our overall picture on the importance of cannibal feeding for the period of food scarcity.

For information on the seasonal and interannual mesozooplankton assemblage in Kiel Bay, see Smetacek V (1983). The annual cycle of Kiel Bight plankton— a long term analysis. *Estuaries* 6:328–328.

#1: A useful addition might be the estimation of the amount of energy invested into reproduction-for-cannibalism, compared to the amount of energy gained from larval cannibalism. Isn't it more profitable to directly invest energy into reserves, rather than reproduction?

Discussion correctly highlights the impact of overfishing and eutrophication on *M. leidyi* proliferation, rather than blaming the comb jelly for everything.

It would be interesting to discuss the work of Augustin et al. 2014 who discuss that *M. leidyi* has a very low reserve capacity, this suggests that “storing” reserves as larvae might thus be a feasible and needed strategy for survival.

REPLY: Thank you for mentioning the excellent Augustine et al. 2014 paper, which we now cite in the Introduction and Discussion. We have added the sentence below in the second Discussion paragraph (lines 171-174) to highlight that further studies are warranted to understand the trade-off between reproduction vs somatic growth as follows:

“However, in order to fully understand the relative costs of adults to invest in reproduction versus the energy they gain from larval cannibalism, it would be necessary to characterize the costs of egg production for field sized adults (~20 mm AO length) as well as the energy larvae gain by predating small size zooplankton.”

Addressing the trade-off between reproductive investment vs. somatic growth would require a lot of additional data and expand our already comprehensive paper. We are convinced that our population data and modeling is sufficient to conclude that investment in reproduction expands the dietary niche of *M. leidyi* and that cannibalism is critical during the period when their basal metabolism is still high. We did consider citing a paper by (Jaspers, Acuña & Brodeur 2015) because the study found that carbon specific eggs production can reach up to 8% of the individuals weight/day, but with the caveat that it is for individuals that are much larger (50-80 mm AO length) than in our study. In this context, we do want to mention that our daily rations are bigger than 8% during some weeks. However, the question of trade-off between reproduction vs somatic growth will remain speculative without a trial with field-relevant sized adults and investigation of the larval prey field.

Recognizing that the cost-benefit of cannibalism remains one of the most interesting questions of our study, we added the following sentence to the final Discussion paragraph (lines 219-223):

*“The extent to which this behaviour confers a fitness advantage on *M. leidyi* adults to invest in reproductive over somatic growth under different conditions requires further parametrization in terms of interspecific predation, exploitation of different prey fields, and costs of egg production (Augustine et al. 2014; Jaspers, Acuña & Brodeur 2015).”*

We thank the Reviewer for these excellent suggestions and hope that they have now improved our Discussion and the Outlook for future research in this regard.

#1: Additionally, this “larval cannibalism” strategy seems to be complementary to the finding that adult *M. leidyi* enhance food availability for their larvae by consuming microzooplankton predators (McNamara et al. 2013).

REPLY: This is an important point. We have now added following sentence to the second Discussion paragraph (lines 161-162):

*“In fact, adult *M. leidyi* may even enhance food availability for their larvae by consuming microzooplankton predators (McNamara, Lonsdale & Cerrato 2013)”*

We thank the Reviewer for this insight and we again believe that it has improved our Discussion.

#1: In methods you describe delta15N values but these are not used in the paper, only the raw 15N values?

REPLY: The Reviewer’s comment is correct because we only report in atom%. We have now removed the sentence describing the ‘delta’ notation and thank them for noticing this.

#1: Fig. 2B shows average size?

REPLY: In panel B, the size plotted is not an average but all data points (6094 individuals). We modified the figure caption to reflect this.

#1: I think your fig. 2 D right axis should be “microzooplankton”?

REPLY: Thank you for noticing this. We have corrected it to microzooplankton.

#1: References

Augustine, S., Jaspers, C., Kooijman, S.A.L.M., Carlotti, F., Poggiale, J.-C., Freitas, V., Van der Veer, H., Van Walraven, L., 2014. Mechanisms behind the metabolic flexibility of an invasive comb jelly. *J. Sea Res.* 94, 156–165.

Jaspers, C., Costello, J., Sutherland, K., Gemmell, B., Lucas, K., Tackett, J., Dodge, K., Colin, S., 2017. Resilience in moving water: Effects of turbulence on the predatory impact of the lobate ctenophore *Mnemiopsis leidyi*: *Mnemiopsis leidyi* feeding in turbulence. *Limnology and Oceanography* 63. <https://doi.org/10.1002/lno.10642>

McNamara, M.E., Lonsdale, D.J., Cerrato, R.M., 2013. Top-down control of mesozooplankton by adult *Mnemiopsis leidyi* influences microplankton abundance and composition enhancing prey conditions for larval ctenophores. *Estuar. Coast. Shelf S.* 133, 2–10.

We have now added all of these References into the main text and thank the Reviewer for drawing our attention to them.

Reviewer #2 (Remarks to the Author):

#2: Javidpour et al. document cannibalism in the ctenophore *Mnemiopsis leidyi* in lab experiments. This is a novel report for the phylum (to my knowledge) and should be of broad interest. More importantly, the authors then make the reasonable conjecture that similar cannibalism happens in the Baltic Sea and that this behavior can explain the ability of the species to survive winter outside its native range with greater energy reserves. This type of experimental work is outside my area of expertise, but they appear and sound compelling. Further, that these experiments provide positive evidence that cannibalism is an important strategy allowing individuals of this ecologically important species to build energy stores to survive winter conditions appears reasonable.

For my taste, however, I think the conclusive language is a bit strong. They have not shown that cannibalism happens in natural conditions, either in the Baltic or in the native range of the species. It seems obvious that this behavior must have evolved in different conditions than those in the Baltic. I'd like to see the authors introduce a bit more discussion on what relevant knowledge is missing in understanding this phenomenon, rather than being so strong in concluding that they have uncovered a "crucial strategy for *M. leidyi* adults to overcome prey scarcity" outside its native habitat. Is the strategy potentially crucial within its native habitat? That said, this knowledge is crucial to understanding the present and potential impact of this invasive species, as the authors rightly point out.

REPLY: We thank the Reviewer for their positive comments in relation to the novel findings and broad interest of our manuscript. We are happy that they think our paper will make a significant contribution to ecological discussions of the phylum and find our experiments and arguments compelling. We have taken their points in relation to conclusive language and have toned this down in our Discussion, highlighting that more parameterization is needed in terms of cost-benefits of cannibalism (lines xx).

We are, however, certain that cannibalism occurs under natural conditions as it is the only possible strategy that the adult population would have sustained in absence of interspecific prey (though, following Reviewer 1, we do now make clear the relationship between our experiment and this conclusion). Moreover, our photographic evidence shows two larvae inside the auricles of an adult collected from the wild (Fig. S1). To make it clear that this was observed in a field sampled adult, we now clarify as (lines 101-106):

"To obtain evidence of cannibalistic behaviour in the natural environment, we photographed field sampled adults within 30 min after sampling. These photographs were taken in the laboratory of Geomar in September 2008. The photographs show two larvae inside the auricles of an adult (see Fig. S1 in the Supplementary Information). We rule out post-capture larval ingestion because the mesh size for capturing the adults was too big for larvae."

#2: Finally, I find the evolutionary discussion of cannibalism, trying to capitalize on the status of Ctenophora as potentially the earliest diverging metazoan lineage to be entirely unconvincing. It is forced and really adds little to the overall understanding of the evolution of cannibalism. This behavior exists within microbial eukaryotes and is widespread across different metazoan lineages. An

alternative explanation is that sometimes ecological conditions favor the evolution of cannibalism and that it is a labile trait. Given that Mnemiopsis is a highly derived ctenophore (see any phylogeny of Ctenophora paper) and one whose biology (being brackish, etc.) is unusual within the phylum, it seems more likely that cannibalism is a derived trait in this instance.

REPLY: The reviewer is correct that Mnemiopsis is a highly derived ctenophore that most likely developed ~300 million years ago, i.e. long after the first metazoans evolved. Hence, it is possible that cannibalism is a derived that a basic trait. However, absence of cannibalism within other members of Ctenophora could simply be explained by the fact that it is difficult to observe, and it is possible that our study is the first of many studies to report cannibalism within this phylum.

To balance the two arguments more evenly, we have removed the original sentence in our Introduction of Mnemiopsis being a model species for understanding basic metazoan traits, and edited the Discussion accordingly (lines 204-209):

“Here, we demonstrate that cannibalism is an important trait for a member of Ctenophora but whether it is a basic or derived metazoan trait remains an open question until more conclusive evidence of cannibalism or lack hereof can be obtained from other members. Hence, more research characterizing the role of cannibalism in a comparative context is essential for understanding the life history strategies of ctenophores and for the evolutionary origins of cannibalism.”

We hope that this now satisfies the Reviewer in this regard.

#2: In summary, this manuscript needs little revision to become published as a paper that will attract wide interest. It documents cannibalism for the first time within Ctenophora, it shows how this behavior likely benefits the species by creating energy stores that allow it to survive winter, and it correctly notes that this information is crucial for marine managers needing to understand its persistence and possible spread to other regions.

REPLY: We thank the Reviewer for their highly positive comments on our paper and their comments that it will attract a lot of wide interest. We also thank them for their highly constructive comments in relation to conclusive language which we now hope to have addressed and improved the balance of our Discussion as a result.

Reviewer #3 (Remarks to the Author):

#3: I believe this paper offers a new, interesting potential explanation for the success that the ctenophore mnemiopsis leidy is having by colonizing new ecosystems. The paper is well written (except for few little misuses of English, likely due to the fact that none of the authors is a native speaker) and integrates several data to support the conclusion that the ctenophore is able to feed on its own larvae and this is the key of its success.

REPLY: We thank the Reviewer for their enthusiastic comments about the interest and novelty of our manuscript. We are glad that they found the paper overall to be well written and are convinced by our experiments and conclusions. While one of the authors, Patrick Roberts, is in fact a native speaker, we have admittedly found some mis-uses (our bad!) which we have now corrected and taken a detailed read through the entire manuscript.

#3: However, this conclusion is based on data collected during only one bloom and adult ctenophores have never been seen while feeding on their larvae nor gut content analyses indicated that larvae were in the gut of adults. Therefore I think the authors should present cannibalism as a potential explanation for the success of the ctenophore and not as "the" explanation. For example, at the beginning of the Discussion they state this paper provides the first "evidence of cannibalism in *M. leidyi*, but I do not see evidence, but data that may support it. I suggest the authors to slightly change the overall tone of the ms in this perspective.

REPLY: As in our responses to Reviewer #2, we concede that our language about cannibalism as written was too conclusive. Nevertheless, as highlighted above, we have now clarified that we found direct evidence for cannibalism in the field, following the discovery of larvae inside the auricles of field collected adults. Since we do have both isotopic and photographic evidence of cannibalism, we maintain that our study shows the first evidence of cannibalism in *M. leidyi*. We have tried to make this clearer and have nevertheless toned down our language in the final paragraph. We hope that this now satisfies the Reviewer.

References

- Augustine, S., Jaspers, C., Kooijman, S.A.L.M., Carlotti, F., Poggiale, J.C., Freitas, V., van der Veer, H. & van Walraven, L. (2014) Mechanisms behind the metabolic flexibility of an invasive comb jelly. *Journal of Sea Research*, **94**, 156-165.
- Fennel, W. (2001) Modeling of copepods with links to circulation models. *Journal of Plankton Research*, **23**, 1217-1232.
- Granhag, L., Møller, L.F. & Hansson, L.J. (2011) Size-specific clearance rates of the ctenophore *Mnemiopsis leidyi* based on in situ gut content analyses. *Journal of Plankton Research*, **33**, 1043-1052.
- Jaspers, C., Acuña, J.L. & Brodeur, R.D. (2015) Interactions of gelatinous zooplankton within marine food webs. *Journal of Plankton Research*, **37**, 985-988.
- Javidpour, J., Molinero, J.C., Peschutter, J. & Sommer, U. (2009) Seasonal changes and population dynamics of the ctenophore *Mnemiopsis leidyi* after its first year of invasion in the Kiel Fjord, Western Baltic Sea. *Biological Invasions*, **11**, 873-882.
- Jones, R.H., Flynn, K.J. & Anderson, T.R. (2002) Effect of food quality on carbon and nitrogen growth efficiency in the copepod *Acartia tonsa*. *Marine Ecology Progress Series*, **235**, 147-156.
- McNamara, M.E., Lonsdale, D.J. & Cerrato, R.M. (2013) Top-down control of mesozooplankton by adult *Mnemiopsis leidyi* influences microplankton abundance and composition enhancing prey conditions for larval ctenophores. *Estuarine, Coastal and Shelf Science*, **133**, 2-10.

REVIEWERS' COMMENTS:

Reviewer #1 (Remarks to the Author):

In their revised manuscript and rebuttal the authors have sufficiently addressed the points raised in my review. The use of a higher value for individual copepod prey carbon content does not appear to change the conclusions of the manuscript. Ideally, estimates of individual carbon content for the meroplankton groups could also be used in estimating mesoplankton prey carbon availability but given the low abundance of these groups in the second half of the study period I doubt that this would change the results much so this is not necessary. In my opinion the paper can be published as is.

Sincerely,

Lodewijk van Walraven

Reviewer #2 (Remarks to the Author):

The authors adequately dealt with my criticisms, as well as those of the other reviewers. I believe it is improved and ready to be published.